# The Anti-Inflammatory Response of *Lavandula luisieri* and *Lavandula pedunculata* Essential Oils

**DOI:** 10.3390/plants11030370

**Published:** 2022-01-29

**Authors:** Monica Zuzarte, Cátia Sousa, Carlos Cavaleiro, Maria Teresa Cruz, Lígia Salgueiro

**Affiliations:** 1Coimbra Institute for Clinical and Biomedical Research (iCBR), Faculty of Medicine, University of Coimbra, 3000-548 Coimbra, Portugal; 2Center for Innovative Biomedicine and Biotechnology (CIBB), University of Coimbra, 3000-548 Coimbra, Portugal; uc45185@uc.pt (C.S.); trosete@ff.uc.pt (M.T.C.); 3Clinical Academic Centre of Coimbra (CACC), 3000-548 Coimbra, Portugal; 4Faculty of Pharmacy, University of Coimbra, 3000-548 Coimbra, Portugal; cavaleir@ff.uc.pt (C.C.); ligia@ff.uc.pt (L.S.); 5Centre for Neuroscience and Cell Biology (CNC), University of Coimbra, 3000-548 Coimbra, Portugal; 6Chemical Process Engineering and Forest Products Research Centre (CIEPQPF), Department of Chemical Engineering, Faculty of Sciences and Technology, University of Coimbra, 3030-790 Coimbra, Portugal

**Keywords:** essential oil, iNOS, interleukin, lavenders, NF-κB

## Abstract

Portuguese lavenders remain undervalued in global markets due to the lack of high-quality end-products and scarcity of scientific-based studies validating their bioactive potential. Moreover, chemical variability is frequent in these species, and can compromise both safety and efficacy. In the present study, the anti-inflammatory potential of *L. luisieri* and *L. pedunculata*, two highly prevalent species in Portugal, was assessed and correlated with their chemical variability. Representative samples with distinct chemical profiles were selected to assess the anti-inflammatory effect on LPS-stimulated macrophages. *L. luisieri* essential oil with low quantities of necrodane derivatives was the most potent at inhibiting NO production. Interestingly, the essential oil was more effective than its main compounds (1,8-cineole and fenchone), assessed alone or in combination. Our results also demonstrated a significant effect of the oil on the expression of the inflammatory proteins (iNOS and pro-IL-1β) and on the NF-κB pathway. Overall, this study highlights the impact of chemical variability on oils’ efficacy by showing distinct effects among the chemotypes. We also identify *L. luisieri* essential oil, with low quantities of necrodane derivatives, as the most promising in the mitigation of the inflammatory response, thus corroborating its traditional uses and paving the way for the development of herbal medicinal products.

## 1. Introduction

In recent years, there has been a growing demand for natural products, with the global market impact of aromatic and medicinal plants increasing mainly in the health sector [1]. Several industries have perceived this trend and are searching for bioactive and biodegradable products that concomitantly present a safe profile to humans and animals [2]. Lamiaceae is among the families that comprise a higher number of aromatic plants with commercial interest [3]. For example, several lavender products, such as essential oils, fresh or dried flowers, and landscape plants, mainly from four *Lavandula* species, namely *L. stoechas*, *L. angustifolia*, *L. latifolia*, and *L.* x *intermedia* (a hybrid between the two last species), are highly appreciated. These species are extremely valuable and recognized in global markets and, in some cases, regulated by international ISO standards [4,5]. Nevertheless, the genus comprises many other species that remain undervalued and have no commercial value. For example, the Portuguese lavenders *L. luisieri* and *L. pedunculata* remain unbranded despite their economic potential, high prevalence, and popularity among local communities [6]. A lack of both scientific-based studies validating their traditional applications and high-quality end-products is partially responsible for this scenario. Indeed, the chemical variability of lavenders is high [7,8,9] and compromises their bioactive potential. Therefore, a well-defined chemical profile and validated bioactive potential are necessary to fulfil the requirements of competitive markets and guarantee efficacy and safety profiles.

*L. luisieri* has been traditionally used as a expectorant, stimulant, spasmolytic, and laxative, for relieving headaches and migraines, and for its disinfectant properties [7,10]. In addition, *L. pedunculata* has been applied to treat bronchitis, cough, asthma, headaches, anxiety, insomnia, strokes, and dyspepsia, and as an analgesic [11,12,13]. Several of these conditions are associated with an inflammatory response and their relief or even treatment may be related to the anti-inflammatory properties of these aromatic plants. Although the anti-inflammatory potential of other lavender species, namely *L. angustifolia*, *L. stoechas*, *L. multifida* and *L. viridis* [14,15,16,17,18], is known, information on *L. luisieri* and *L. pedunculata* essential oils is sparse. Only two studies were carried out on *L. luisieri* essential oils, one showing its effect on primary human chondrocytes and the intestinal cell line [19], and another highlighting its potential to reduce cytokines and chemokines in human acute monocytic leukemia cells (THP-1) [20]. Taking this into account, and the fact that chronic inflammation has been associated with aging and related diseases [21,22,23] that greatly impact the quality of life, it is imperative to search for effective and safe anti-inflammatory therapeutic strategies. Therefore, using an *in vitro* model of lipopolysaccharide (LPS)-stimulated macrophages, the present study aimed to evaluate the anti-inflammatory potential of these essential oils with distinct chemical profiles. For the most promising oil, the effect of the main compounds, alone or in combination, was also investigated, and its mechanism of action underlying the pharmacological effects, namely the nuclear factor kappa B (NF-κB) signaling pathway, was further explored. Our results demonstrate relevant anti-inflammatory properties for the chemotype of *L. luisieri* essential oil with low quantities of necrodane derivatives, thus paving the way for the development of new plant-based anti-inflammatory therapeutics.

## 2. Results

### 2.1. Essential Oil Characterization

Samples of *L. luisieri* and *L. pedunculata* were collected throughout Portugal and their essential oils were further analyzed. The main chemical compounds present in the different samples allowed the identification of two chemotypes for *L. luisieri* and three for *L. pedunculata*. *L. luisieri* chemotypes were distinguished by the amounts of necrodane derivatives, with some samples presenting low concentrations of these compounds (6.9 ± 4.6%) and others showing high amounts (24.9 ± 2.4%). For *L. pedunculata*, differences occurred in the concentrations of three main compounds, with samples rich in fenchone (36.0% ± 12.9), 1,8-cineol (32.9% ± 8.3) or camphor (37.5% ± 8.1). The main compounds of representative samples of the identified chemotypes are listed in Table 1. Overall, the essential oils from both species were obtained with a yield of 0.8–1.10% and were characterized by high contents of oxygen-containing monoterpenes (54.9–80.3%). Regarding *L. luisieri*, the concentrations of 1,8-cineole, fenchone and necrodane derivatives differed among samples, with sample A1 characterized by low amounts of necrodane derivatives and high amounts of 1,8-cineole and fenchone, whereas in sample A2, necrodane derivatives occurred in higher amounts. For *L. pedunculata*, three main compounds (1,8-cineole, fenchone and camphor) were always identified, but their concentrations differed significantly between samples, with B1 rich in 1,8-cineole, B2 in fenchone, and B3 in camphor.

### 2.2. Nitric Oxide Scavenging Potential of Lavandula luisieri and Lavandula pedunculata Essential Oils

The antioxidant potential of the essential oils towards reactive nitrogen species (RNS) was assessed using an *in chemico* nitrite-scavenging assay. Overall, the essential oils were ineffective and no nitric oxide (NO) scavenging effect was observed for all the tested concentrations (Figure 1). Indeed, nitrite values remained very similar to those of the NO donor, SNAP, in both *L. luisieri* (Figure 1A,B) and *L. pedunculata* (Figure 1C–E) chemotypes.

### 2.3. Effect of Lavandula luisieri and Lavandula pedunculata Essential Oils on Macrophages’ Viability

To evaluate the anti-inflammatory activity of the essential oils, we first evaluated their effect on cell viability of murine macrophages stimulated with LPS (Figure 2), in order to select non-toxic concentrations. For all the essential oils, a range of concentrations varying from 0.08 to 0.64 μL/mL was tested. Overall, the essential oils were devoid of toxicity, except for the highest concentration of *L. luisieri* essential oil rich in necrodane derivatives (sample A2), which decreased cell viability by more than 70% compared to LPS-treated cells (Figure 2B). The vehicle, DMSO, did not interfere with macrophages’ viability compared to control cells (Appendix A).

### 2.4. Effect of Lavandula luisieri and Lavandula pedunculata Essential Oils on NO Production

Using an *in vitro* model of LPS-stimulated macrophages referred to previously, the non-toxic concentrations of the essential oils determined in the previous section were used to assess the effect of the oils with distinct chemical profiles on LPS-induced NO production. Macrophages produce residual levels of nitrites (stable metabolite of NO). However, when stimulated with LPS, NO production increases considerably when compared to control. *L. luisieri* essential oils having both low amounts of necrodane derivatives (sample A1) and high amounts of necrodane derivatives (sample A2), significantly decreased LPS-induced NO production in a concentration-dependent manner (Figure 3A,B, respectively). However, sample A1 was much more effective in decreasing NO production (Figure 3A) than sample A2 (Figure 3B) at the concentrations tested without compromising cell viability (Figure 2A). Regarding *L. pedunculata* essential oils, the samples with high amounts of 1,8-cineole (B1; Figure 3C) and high amounts of camphor (B3; Figure 3E) were able to inhibit NO production induced by LPS, in a concentration-dependent manner, with the latter being more active at the lower concentration tested. By comparison, the sample with high amounts of fenchone (B2; Figure 3D) significantly inhibited NO production, but only at the higher concentrations tested (0.32 and 0.64 µL/mL). The vehicle, DMSO, did not interfere with LPS-induced NO production when compared to LPS-treated cells (Appendix A).

Overall, regarding the ability of the essential oils to inhibit NO production, *L. luisieri* essential oil with low amounts of necrodane derivatives (sample A1) was the most potent (IC_50_ = 0.07 µL/mL; concentration range within the 95% confidence interval: 0.06 to 0.08 µL/mL).

### 2.5. Effect of 1,8-Cineole and Fenchone on NO Production

As *L. luisieri* essential oil with low amounts of necrodane derivatives (sample A1) was the most potent in inhibiting NO production induced by LPS in macrophages, its main compounds (1,8-cineole and fenchone) were also assessed to determine if the observed effect was due to these compounds. Our results showed that 1,8-cineole was not able to decrease LPS-induced NO production at non-toxic concentrations (Figure 4B,C). By comparison, fenchone significantly decreased NO production at the highest non-toxic concentrations tested (Figure 4E,F). Strikingly, when both compounds were combined in the same proportions as they occurred in the essential oil (33.9% 1,8-cineole and 18.2% fenchone), no effect was observed in LPS-induced NO production at non-toxic concentrations (Figure 4G,H). Notably, the effective concentrations for 1,8-cineole, fenchone, and the mixture were higher (≥0.64 μL/mL) in comparison to those tested for *L. luisieri* essential oil (sample A1), which suggests that other compounds are responsible for the bioactive potential of this essential oil, or that synergisms may occur between compounds.

### 2.6. Effect of Lavandula luisieri Essential on the Expression of Inflammatory Mediators

As *L. luisieri* rich in 1,8-cineole and fenchone was the most potent essential oil tested and was more active than its major compounds assessed alone or in combination, we sought to confirm its anti-inflammatory potential. Thus, the effect on the levels of inflammatory proteins, namely inducible nitric oxide synthase (iNOS), ciclooxigenase-2 (COX-2), and the immature form of interleukin—1β (pro-IL-1β), was assessed.

In untreated cells (control) and in cells treated with the essential oil, without LPS stimulation, iNOS, COX-2 and pro-IL-1β were not detected or were slightly expressed (Figure 5). However, after LPS stimulation for 24 h, the levels of these mediators significantly increased (Figure 5). Nevertheless, when macrophages were concomitantly treated with LPS and *L. luisieri* essential oil (0.32 µL/mL), iNOS levels significantly decreased compared to LPS-treated cells (Figure 5A,B), corroborating the previous results demonstrating NO inhibition (Figure 3A). However, *L. luisieri* essential oil did not interfere with LPS-induced COX-2 levels (Figure 5A,C). Interestingly, the essential oil significantly increased the levels of pro-IL-1β induced by LPS (Figure 5A,D), thus suggesting that the conversion of pro-IL-1β into its mature form IL-1β may be inhibited.

### 2.7. Effect of L. luisieri Essential Oil on NF-κB/p65 Nuclear Translocation

A complex network of intracellular signaling pathways and transcription factors tightly regulate the expression of pro-inflammatory molecules. Among the signaling cascades, the NF-κB assumes a decisive role during inflammation. In basal conditions, NF-κB is retained in the cytoplasm by its inhibitor, IκB. Upon a specific stimulus, phosphorylation and, consequently, proteasomal degradation of IκB occurs. Then, free NF-κB dimers, composed of p65 and p50, rapidly translocate into the nucleus where transcription of target genes, such as *NOS2* and *COX-2*, is promoted [25]. To assess the effect of the oil on this signaling pathway, immunocytochemistry for the NF-κB subunit, p65, was performed. As expected, in untreated cells, immunoreactivity for NF-κB/p65 was exclusively located in the cytoplasm (Figure 6). However, after LPS treatment, NF-κB/p65 immunoreactivity was mostly observed in the nucleus (Figure 6). Treatment with *L. luisieri* essential oil (0.32 µL/mL) clearly impaired nuclear translocation of NF-κB/p65, as shown in Figure 6. Moreover, pre-treatment with MG-132 (10 µM), a synthetic proteasome inhibitor peptide, fully inhibited NF-κB/p65 nuclear translocation, as immunoreactivity is only visible in the cytoplasm (Figure 6). The vehicle (DMSO) did not interfere with NF-κB/p65 nuclear translocation when compared to control cells (data not shown). These results suggest that *L. luisieri* essential oil decreased iNOS protein levels by interfering with NF-κB/p65 nuclear translocation.

## 3. Discussion

The Lamiaceae family comprises a high number of aromatic and medicinal plants with commercial interest. Among these species, lavenders stand out due to their high essential oil yield and pleasant odor, which are both important features for industrial applications. Indeed, the typical lavender scent is globally recognized and is included in many home and bath-care products. Lavender essential oil is considered a valuable raw-material in cosmetics and perfumes, and for flavoring [3,26]. Although the *Lavandula* genus comprises 39 wild essential oil bearing species, only four are commercially valued [3]. Nevertheless, some of the remaining species have economic potential mainly due to their bioactive properties. In fact, *L. luisieri* and *L. pedunculata* are widely used in traditional medicine by local communities, but scientific information validating these uses remains sparse. Moreover, local communities use these plants but are unaware of their chemical variability, which can compromise both safety and efficacy. Indeed, *L. luisieri* essential oils have a peculiar chemical composition due to the presence of necrodane derivatives that do not occur in other species [7,27,28,29,30,31] and, although these compounds are always present, significant variations among samples can occur [7]. Therefore, in the present work, two samples, one with low and another with high amounts of these compounds, were studied. Regarding *L. pedunculata*, the chemical composition of the essential oils from Portugal showed similarities with that reported for *L. stoechas* from other Mediterranean countries, namely from Spain [27], Greece [32] and Turkey [33]. Although *L. pedunculata* and *L. stoechas* are morphologically distinct, their essential oils present a similar chemical profile. This is interesting because *L. stoechas* is one of the four species with commercial value, thus suggesting that *L. pedunculata* can be used for the same purposes. Both species present the same major compounds, namely 1,8-cineole, fenchone and camphor, and the amount of these compounds can vary considerably. Therefore, three representative samples, with high amounts of each one of these compounds, were also selected.

As *L. luisieri* and *L. pedunculata* are used in traditional medicine to treat several conditions that share an inflammatory component, the present study aimed to validate the anti-inflammatory potential of these species considering their high chemical variability [7,8], which can affect oil quality and compromise both safety and bioactivity profiles. Inflammation is considered an important mechanism that maintains homeostasis, regardless of whether the insult is exogenous, as in the case of infections, or endogenous, as occurs, for instance, in metabolic disorders [34]. During this process, the first line of defense is provided by macrophages, which in the presence of a Toll-like receptor agonist such as microbial LPS, produce several pro-inflammatory mediators, including NO, COX-2, prostaglandins and cytokines, such as IL-1β [35]. Under normal conditions, the release of these pro-inflammatory mediators for a short period of time is of utmost importance, and aims to eradicate the harmful stimuli. However, the abnormal and sustained production of these mediators may result in damage to the host tissue, leading to a vicious circle that, if maintained over a long period, may evolve to chronic inflammation that has been associated with several aged-related diseases. One of the most important players in this process is the transcription factor NF-κB. Therefore, inhibition of NF-κB transcriptional activity, and its downstream mediators, represents a valuable therapeutic strategy for intervention in inflammation-based pathologies. In this context, the knowledge of phytochemicals’ molecular mechanisms is a good strategy in the search for novel anti-inflammatory compounds. Taking into consideration the above, in this work we performed an initial screening assay using an *in vitro* inflammation model, specifically macrophages stimulated with LPS. Under these experimental conditions, LPS activates the pro-inflammatory transcription factor NF-κB/p65, which rapidly translocates into the nucleus to trigger the transcription of its target genes, such as *NOS2*, *COX-2* and *Il1*. Overall, the results achieved demonstrated that *L. luisieri* and *L. pedunculata* essential oils display anti-inflammatory potential, nonetheless presenting distinct efficacies and safety profiles depending on their chemical composition. In fact, *L. luisieri* essential oil with low amounts of necrodane derivatives was the most potent in mitigating the anti-inflammatory response, as assessed by the inhibition of NO production evoked by LPS. Interestingly, the main compounds of the oil, namely 1,8-cineole and fenchone, were not the only compounds responsible for the observed effect, because they were much less effective than the oil, both alone and in combination, thus indicating that other minor compounds are responsible for the effect of the oil, or that synergistic effects among compounds may occur. In addition, the results achieved demonstrated that the observed anti-inflammatory effect was not related to a scavenging effect of the oil; therefore, the putative mechanism underlying the anti-inflammatory potential was further explored by addressing the effect of the oil on the expression of relevant inflammatory proteins. Our results clearly demonstrated that the anti-inflammatory potential ascribed to *L. luisieri* essential oil (with low amounts of necrodane derivatives) was achieved through inhibition of nuclear translocation of NF-κB/p65, which consequently leads to a decrease in iNOS protein levels and NO production. Interestingly, the essential oil also significantly increased the LPS induced expression of pro-IL-1β, the precursor of the potent pro-inflammatory cytokine IL-1β. It would be expected that pro-IL-1β levels were decreased because the essential oil compromised the nuclear translocation of NF-κB/p65. Nevertheless, we hypothesize that the observed accumulation indicates that pro-IL-1β is not being converted into IL-1β, thus suggesting an effect of the oil on the inflammasome, a proteolytic complex responsible for the maturation and secretion of the inflammatory and pyrogenic IL-1β. However, further studies should be performed to prove this hypothesis.

The results presented herein were corroborated by previous studies describing the anti-inflammatory potential of *L. luisieri* essential oil in both primary human chondrocytes and in an intestinal cell line [19], and in THP-1 cells [20]. The first study showed a decrease in iNOS levels and, consequently, in NO production, and reported that the effect was due to the inhibition of IκB phosphorylation and degradation, thus blocking NF-κB activation [19]. The latter pointed out a reduction in the cytokine tumor necrosis factor-α (TNFα) and the chemokine (C-C motif) ligand 2 (CCL2), following LPS stimulation. Regarding other lavender species, studies on *L. angustifolia* have frequently reported inhibitory effects on carrageenan-induced paw oedema [15], croton oil-induced ear oedema, and dextran-induced paw oedema models [36]. Moreover, decreases in the levels of TNF-α and IL-1β, and increases in IL-10, were also shown in a rat model of myocardial infarction [37]. *L. stoechas* also showed strong lipoxygenase inhibitory effects [14] and our group previously reported the ability of *L. viridis* essential oil to inhibit LPS-induced NO production through down-modulation of NF-κB-dependent *Nos2* transcription and, consequently, iNOS protein expression, in addition to a decrease in proteasomal activity, inhibition of *Il1b* and *Il6* transcription, and down-regulation of COX-2 levels [18]. Overall, these previous studies highlight the anti-inflammatory potential of lavender species, and the current study contributes by providing additional scientific information regarding less recognized species.

To conclude, our results indicate potent anti-inflammatory activity of *L. luisieri* essential oil via down-modulation of the NF-κB pathway, with significant inhibitions in major inflammatory mediators. In addition, the importance of well-defined chemical products is highlighted, because chemical variability compromises essential oils’ quality and, consequently, their safety profile and efficacy. Moreover, our findings corroborate the traditional uses ascribed to these species and indicate that *L. luisieri*, with low amounts of necrodane derivatives, is the most promising for the development of safe anti-inflammatory agents, thus contributing to its industrial valorization.

## 4. Materials and Methods

### 4.1. Plant Material

Flowering parts of representative samples of *L. luisieri* (Rozeira) Rivas Mart. and *L. pedunculata* (Mill.) Cav. (Table 2) were collected. Voucher specimens were included in the Herbarium of the University of Coimbra (COI). Species authenticity was confirmed by Dr. Jorge Paiva, a taxonomist at the University of Coimbra, and plant names were checked at http://www.theplantlist.org accessed on 20 December 2021.

### 4.2. Essential Oil Isolation and Analysis

Flowering aerial parts of *L. luisieri* and *L. pedunculata* were submitted to hydrodistillation for 3 h in a Clevenger-type apparatus [38]. Essential oils were analyzed by gas chromatography (GC) on a Hewlett Packard 6890 gas chromatograph (Agilent Technologies, Palo Alto, CA, USA) with a HP GC ChemStation Rev. A.05.04 data handling system, equipped with a single injector and two flame ionization detectors (FID). A graphpak divider (Agilent Technologies, Part Number 5021-7148) was used for simultaneous sampling in two Supelco (Supelco Inc., Bellefont, PA, USA) fused silica capillary columns with different stationary phases: SPB-1 (polydimethylsiloxane; 30 m × 0.20 mm i.d., film thickness 0.20 µm), and SupelcoWax 10 (polyethylene glycol; 30 m × 0.20 mm i.d., film thickness 0.20 µm). The oven temperature program was: 70–220 °C (3 °C/min), 220 °C (15 min); injector temperature: 250 °C; detector carrier gas: He, adjusted to a linear velocity of 30 cm/s; splitting ratio 1:40; detector temperature: 250 °C. Gas chromatography-mass spectrometry (GC/MS) analyses were performed on a Hewlett Packard 6890 gas chromatograph fitted with a HP1 fused silica column (polydimethylsiloxane; 30 m × 0.25 mm i.d., film thickness 0.25 µm), interfaced with an Hewlett Packard Mass Selective Detector 5973 (Agilent Technologies, Palo Alto, CA, USA) operated by HP Enhanced ChemStation software, version A.03.00. GC parameters were as above; interface temperature: 250 °C; MS source temperature: 230 °C; MS quadrupole temperature: 150 °C; ionization energy: 70 eV; ionization current: 60 µA; scan range: 35–350 u; scans/s: 4.51. Retention indices (RIs) and mass spectra were used to identify volatile compounds. RIs were calculated by linear interpolation relative to retention times of a series of n-alkanes, and compared with those of authenticated samples from the database of the Laboratory of Pharmacognosy, Faculty of Pharmacy, University of Coimbra. Mass spectra were compared with reference spectra from a home-made library or from literature data [39,40]. Relative amounts of individual components were calculated based on GC peak areas without FID response factor correction.

### 4.3. Nitric Oxide Scavenging Potential

The essential oils’ NO scavenging potential was evaluated using S-nitroso-N-acetyl-DL-penicillamine (SNAP) as a NO donor. NO production was measured using Griess reaction. Briefly, in 48-well plates, 300 µL of culture medium alone (control) or with different concentrations of the essential oils and 300 µM of SNAP were incubated for 3 h at 37 °C. After this period, equal volumes of the supernatants and Griess reagent [1% (*w*/*v*) sulphanilamide in 2.5% (*v*/*v*) phosphoric acid and 0.1% (*w*/*v*) naphthylethylenediamine dihydrochloride] were mixed and incubated in the dark, for 30 min, at room temperature. The absorbance was then read at 550 nm using a Biotek Synergy HT plate reader (Biotek, CA, USA). The concentration of nitrites accumulated in supernatants was calculated by interpolation of the absorbance of each sample in a standard curve of sodium nitrite. All experiments were performed in triplicate.

### 4.4. Cell Culture and Treatments

The mouse leukemic macrophage cell line, RAW 264.7 (ATCC—TIB-71), was cultured in Dulbecco’s Modified Eagle Medium supplemented with 10% (*v*/*v*) non-inactivated Fetal Bovine Serum, 100 μg/mL streptomycin and 100 U/mL penicillin. The cell line was maintained at 37 °C, in a humidified atmosphere of 95% air and 5% CO_2_, and used after reaching 80–90% confluence. Cells were plated at 3 × 10^5^ cells/well and left to stabilize for 12 h.

For cell treatments, essential oils, 1,8-cineol (extra pure; Merck, Darmstadt, Germany) and fenchone (99.9% purity; Fluka AG, Buchs, Switzerland) were initially diluted in dimethyl sulfoxide (DMSO; Sigma-Aldrich Co., St. Louis, Mo, USA) at a 1:1 proportion and, then, serial dilutions were performed in culture medium. MG-132 (Z-Leu-Leu-Leu-CHO, Boston Biochem, Cambridge, MA, USA) was also dissolved in DMSO (Sigma-Aldrich Co). In both cases, the final concentration of DMSO did not exceed 0.1% (*v*/*v*). LPS from *Escherichia coli* 026:B6 (Sigma-Aldrich Co.) was dissolved in phosphate buffered saline (PBS). The concentrations of the essential oils, 1,8-cineol, fenchone, MG-132 and LPS, and the experimental treatment periods, are indicated in figures and/or figure legends.

### 4.5. Cell Viability

Assessment of cell viability was performed using the 3-(4,5-dimethylthiazol-2-yl)-2,5-diphenyl tetrazolium bromide (MTT) assay. After the treatment period, a MTT solution (final concentration 0.5 mg/mL; Sigma-Aldrich Co.) was added to each well and further incubated at 37 °C for 15 min in a humidified atmosphere of 95% air and 5% CO_2_. Acidified isopropanol (0.04 N HCl in isopropanol) was added to each cell to dissolve formazan crystals and quantification was carried out using an Biotek Synergy HT plate reader (Biotek) at 570 and 620 nm (reference wavelength). A cell-free control was performed in order to exclude non-specific effects of the essential oils, 1,8-cineol and fenchone on MTT (data not shown). All experiments were performed in triplicate.

### 4.6. Nitric Oxide Production

NO production was measured as the amount of nitrite accumulation in the culture supernatants using the Griess reaction described in Section 4.3.

### 4.7. Western Blotting

Total cell extracts were obtained using RIPA buffer (50 mM Tris-HCL, pH 8.0, 1% Nonidet P-40, 150 mM NaCl, 0.5% sodium deoxycholate, 0.1% sodium dodecyl sulphate (SDS) and 2 mM ethylenediaminetetraacetic acid) freshly supplemented with 1 mM dithiothreitol (DTT), protease (Complete Mini, Roche Diagnostics, Mannheim, Germany) and phosphatase (PhosSTOP, Roche Diagnostics, Mannheim, Germany) inhibitor cocktails. The cell lysates were then sonicated (4 times, 40 µm peak to peak) in a Vibra Cell sonicator (Sonica and Material INC) and centrifuged for 10 min at 4 °C to remove nuclei and cell debris. Protein concentration in the total lysates was determined using the bicinchoninic acid kit (Sigma-Aldrich Co.) and cell lysates were denatured in sample buffer [0.125 mM Tris pH 6.8, 2% (*w*/*v*) SDS, 100 mM DTT, 10% glycerol and bromophenol blue] at 95 °C for 5 min. Equal amounts of proteins were then separated by SDS-PAGE and electrotransferred onto PDVF membranes. After blocking with 5% (*w*/*v*) non-fat milk in Tris-Buffered saline (TBS)-Tween 20 (0.1%) for 1 h, membranes were incubated overnight at 4 °C with the primary antibodies indicated in Table 3 and then with anti-rabbit (dilution 1:20,000; NIF1317, GE Healthcare, Chalfont St. Giles, UK) or anti-mouse (dilution 1:20,000; NIF1316, GE Healthcare, Chalfont St. Giles, UK) alkaline phosphatase-conjugated secondary antibodies. Immune complexes were detected with Enhanced ChemiFluorescent reagent (GE Healthcare) in the Typhoon^TM^ FLA 9000 imaging system. Mouse anti-actin was used as loading control. Image analysis was performed with Image Quant TL software.

### 4.8. Immunocytochemistry

To evaluate NF-κB/p65 nuclear translocation, immunocytochemistry was performed as previously described [41]. Briefly, at the end of the treatment period, cells were washed with ice-cold PBS pH = 7.4 and, then, fixed in 4% paraformaldehyde at room temperature, for 15 min. After fixing, cells were blocked with 5% Goat Serum, 0.3% Triton in PBS, pH = 7.4 for 1 h at room temperature. Then, slides were incubated with a rabbit monoclonal anti-NF-κB p65 (D14E12) XP^®^ antibody (dilution 1:400; #8242, Cell Signaling Technology, Inc.) in 1% Bovine Serum Albumin in PBS (pH = 7.4) overnight at 4 °C and then incubated with anti-rabbit IgG (H+L) CF^TM^488A (dilution 1:400; SAB4600165, Biothium, Inc., Fremont, CA, USA) for 1 h at room temperature in the dark. The cells were counterstained with DAPI (dilution 1:1,000; Sigma-Aldrich D9542) to stain the nuclei. Specificity was confirmed in negative controls set up by omitting the primary antibody. Fluorescence images were obtained in a widefield fluorescence microscope (Axio Observer.Z1; Carl Zeiss, Germany) and images were acquired using Zen Black 2010 software.

### 4.9. Statistical Analysis

Results are expressed as mean ± SEM. Statistical analysis was performed with GraphPad Prism version 9 (GraphPad Software, San Diego, CA, USA), using a two-sided unpaired t-test for comparisons between a control and treated group or using one-way ANOVA with Dunnett’s post-test for multiple comparisons to a control group. Results were considered statistically significant at *p* < 0.05.

## Figures and Tables

**Figure 1 plants-11-00370-f001:**
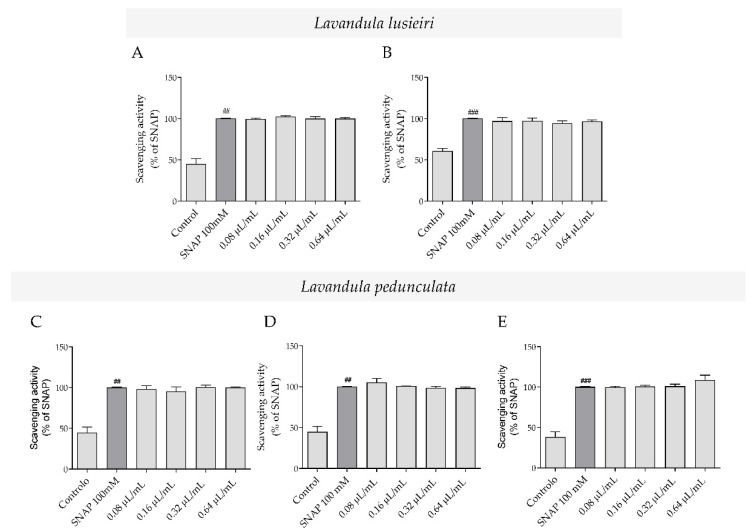
NO scavenging potential of *Lavandula* spp. essential oils. Effect of *L. luisieri* essential oil with (**A**) low (sample A1) or (**B**) high (sample A2) amounts of necrodane derivatives and *L. pedunculata* rich in (**C**) 1,8-cineole (sample B1), (**D**) fenchone (sample B2) and (**E**) camphor (sample B3). Different concentrations of essential oils (0.08–0.64 μL/mL) were incubated with the NO donor, SNAP (100 mM), in culture medium for 3 h. Results are expressed as percentage of NO release triggered by SNAP (positive control, grey bars). Each value represents the mean ± SEM of three experiments, performed in duplicate ((## *p* < 0.01, ### *p* < 0.001, compared to control).

**Figure 2 plants-11-00370-f002:**
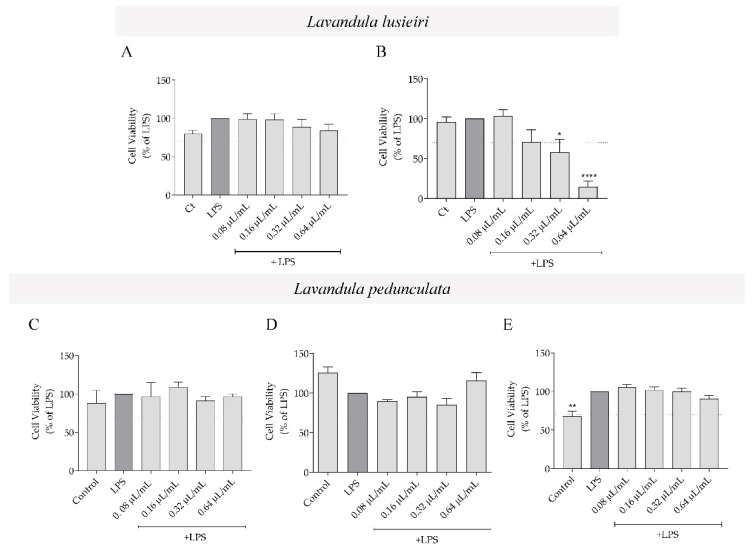
The effect of *Lavandula* spp. essential oils on cell viability. Effect of *L. luisieri* essential oil with (**A**) low (sample A1) or (**B**) high (sample A2) amounts of necrodane derivatives and *L. pedunculata* rich in (**C**) 1,8-cineole (sample B1), (**D**) fenchone (sample B2) and (**E**) camphor (sample B3) on macrophages’ viability. Cells were maintained in culture medium (control), or incubated with 1 μg/mL LPS or with LPS in the presence of different concentrations of the oil (0.08–0.64 μL/mL), for 24 h. Results are expressed as percentage of MTT reduction by cells treated with LPS. Each value represents the mean ± SEM of three experiments, performed in duplicate (* *p* < 0.05, ** *p* < 0.01 and **** *p* < 0.0001, compared to LPS). The dotted line represents the threshold (70% of maximal viability) below which cytotoxicity is recognized, in agreement with standard ISO 10993-5 [24].

**Figure 3 plants-11-00370-f003:**
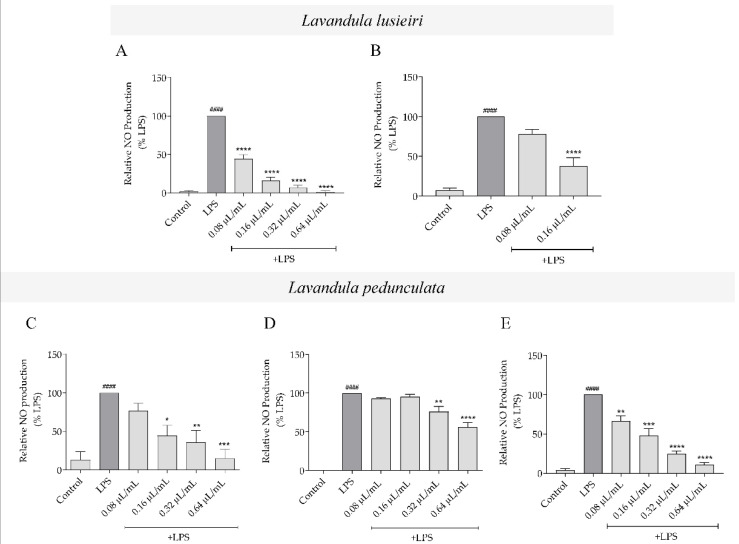
Inhibition of NO production by *Lavandula* spp. essential oils. Effect of *L. luisieri* essential oil with (**A**) low (sample A1) or (**B**) high (sample A2) amounts of necrodane derivatives and *L. pedunculata* rich in (**C**) 1,8-cineole (sample B1), (**D**) fenchone (sample B2) and (**E**) camphor (sample B3) on NO production induced by LPS in macrophages. Cells were maintained in culture medium (control), or incubated with 1 μg/mL LPS, or with LPS in the presence of different concentrations of the oil (0.08–0.64 μL/mL), for 24 h. Nitrite concentration was determined from a sodium nitrite standard curve and the results are expressed as a percentage of NO production by cells treated with LPS. Each value represents the mean ± SEM of three independent experiments, performed in duplicate (#### *p* < 0.0001, compared to control; * *p* < 0.05, ** *p* < 0.01 and *** *p* < 0.001 and **** *p* < 0.0001, compared to LPS).

**Figure 4 plants-11-00370-f004:**
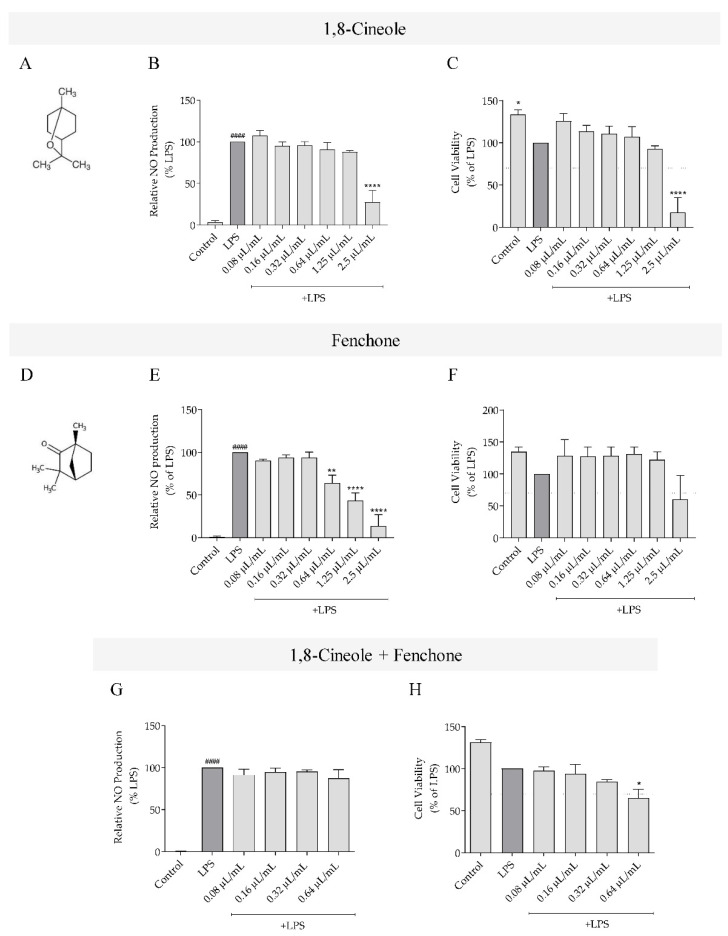
Effect of 1,8-cineole, fenchone, and their mixture on NO production. Chemical structure of (**A**) 1,8-cineole and (**D**) fenchone. Effect of 1,8-cineole (**B**,**C**), fenchone (**E**,**F**) and 1,8-cineole combined with fenchone (**G**,**H**) on NO production and cell viability, respectively. Cells were maintained in culture medium (control), or incubated with 1 μg/mL LPS, or with LPS in the presence of different concentrations of the compounds (0.08– 2.5 μL/mL), for 24 h. Nitrite concentration was determined from a sodium nitrite standard curve and the results are expressed as a percentage of NO production by cells treated with LPS. Cell viability results are expressed as percentage of MTT reduction by cells treated with LPS. Each value represents the mean ± SEM of three experiments, performed in duplicate (#### *p* < 0.0001, compared to control; * *p* < 0.05, ** *p* < 0.01 and **** *p* < 0.0001, compared to LPS). The dotted line in cell viability graphs represents the threshold (70% of maximal viability) below which cytotoxicity is recognized, in agreement with standard ISO 10993-5 [24].

**Figure 5 plants-11-00370-f005:**
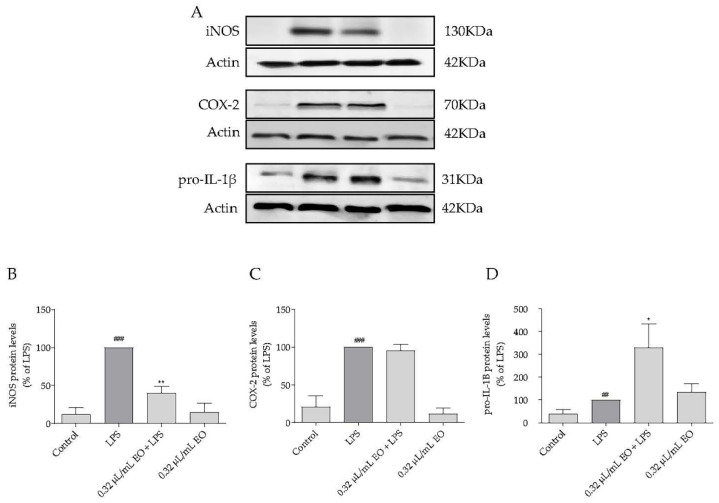
Inhibitory effect of *Lavandula luisieri* essential oil (EO) on LPS-induced inflammatory proteins. (**A**) Representative Western blots of iNOS, COX-2 and pro-IL-1β. Quantification of (**B**) iNOS, (**C**) COX-2 and (**D**) pro-IL-1β protein levels. Cells were maintained in culture medium (control), or incubated with 1 μg/mL LPS, or incubated with the essential oil (0.32 μL/mL) alone or simultaneously with 1 μg/mL LPS, for 24 h. Results are expressed as percentage of protein levels relative to LPS. Each value represents the mean ± SEM from at least 3 experiments (## *p* < 0.01; ### *p* < 0.001, compared to control; * *p* < 0.05, ** *p* < 0.01, compared to LPS). Each value represents the mean ± SEM from three experiments.

**Figure 6 plants-11-00370-f006:**
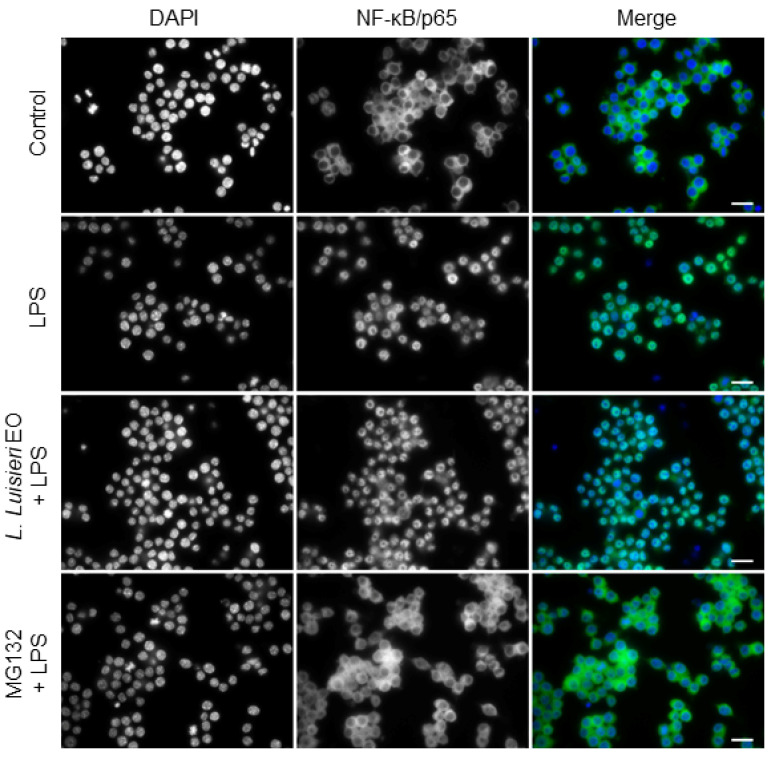
Inhibitory effect of *Lavandula luisieri* essential oil (EO) on NF-κB/p65 nuclear translocation. Macrophages were incubated on μ-slides and pre-treated with MG-132 (10 µM) for 1 h and, then, treated with 1 µg/mL LPS for 20 min, alone or in combination with the essential oil (0.32 µL/mL) or MG-132. Control cells (Ctrl) were left untreated. Immunofluorescence staining of NF-κB/p65 (green) was performed as detailed in the Materials and Methods section and DAPI (blue) was used as a counter stain. Representative images of each condition are shown. Scale bar: 20 µm.

**Table 1 plants-11-00370-t001:** Main compounds of *Lavandula luisieri* and *Lavandula pedunculata* essential oils.

			*Lavandula luisieri*		*Lavandula pedunculata*		
RI SPB-1	RI SW 10	Compound	A1	A2	B1	B2	B3
930	1030	α-pinene	1.1	3.2	2.5	3.9	3.8
942	1075	camphene	0.5	-	0.8	1.8	6.1
969	1116	β-pinene	0.7	1.6	9.0	0.2	1.4
1020	1215	1,8-cineole	33.9	6.4	34.3	12	25.1
1065	1400	fenchone	18.2	-	7.6	49.5	6.2
1082	1542	linalool	3.0	6.2	3.8	2.4	1.2
1118	1514	camphor	2.2	2.5	9.9	15	34.0
1121	1645	*cis*-verbenol	-	0.2	2.8	0.3	0.2
1125	1669	*trans*-verbenol	-	-	1.1	0.4	2
1130	1657	*trans*-*α*-necrodol	4.5	7.1	-	-	-
1146	1692	borneol	-	-	3.4	0.3	0.6
1154	1712	1,1,2,3-tetramethyl-4-hidroximethyl-2-cyclopentane	1.1	2.0	-	-	-
1159	1645	2,3,4,4-tetramethyl-5-methylene-cyclopent-2-enone	0.3	2.8	-	-	-
1165	1621	myrtenal	-	0.3	2.4	0.4	0.8
1265	1590	*trans*-*α*-necrodyl acetate	3.2	17.4	-	-	-
1269	1602	lavandulyl acetate	2.2	7.6	-	-	-
1269	1657	lyratyl acetate	0.3	2.4	-	-	-
1301	1683	myrtenyl acetate	2.0	-	-	-	-
1571	2068	viridiflorol	1.4	2.1	-	-	-
1628	2218	α-cadinol	-	0.7	3.1	0.2	0.2
Monoterpene hydrocarbons			2.3	4.5	12.3	5.9	11.3
Oxygen containing monoterpenes (including necrodane derivatives)			72.3	54.9	65.3	80.3	70.1
Oxygen containing sesquiterpenes			1.4	2.8	3.1	0.2	0.2

Compounds listed in order of elution from the SPB-1 column. RI SPB-1: GC-retention indices relative to C9–C23 n-alkanes on the SPB-1 column. RI SW 10: GC-retention indices relative to C9–C23 n-alkanes on the SupelcoWax-10 column. A1 and A2—representative samples of *L. luisieri* chemotypes; B1–B3—representative samples of *L. pedunculata* chemotypes. (-)—not detected.

**Table 2 plants-11-00370-t002:** Site of collection of *Lavandula* spp.

Species	Region	Site of Collection	Sample
*L. luisieri*	Coimbra	Piódão	A1
	Algarve	Cabo de São Vicente	A2
*L. pedunculata*	Guarda	Celorico da Beira	B1
	Bragança	Serra da Nogueira	B2
	Coimbra	Foz de Arouce	B3

A1 and A2—representaive samples of *L. lusieri*; B1–B3—representative samples of *L. pedunculata*.

**Table 3 plants-11-00370-t003:** List of primary antibodies used in Western blot assays.

Protein	Source	Clonality	Dilution	Supplier	Catalogue Number
COX-2	rabbit	polyclonal	1:10,000	Abcam, Cambridge, UK	ab6665
IL-1β	rabbit	polyclonal	1:1000	Abcam	ab9722
iNOS	mouse	monoclonal	1:1000	R&D Systems, Minneapolis, MN, USA	MAB9502
Actin	mouse	monoclonal	1:20,000	Sigma-Aldrich Co.	MAB1501

## Data Availability

The data presented in this study are available on request from the corresponding author.

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
