# Peer review of "The Anti-Inflammatory Response of Lavandula luisieri and Lavandula pedunculata Essential Oils"

_plants, 2022, doi:10.3390/plants11030370_

Round 1
Reviewer 1 Report
In the paper The anti-inflammatory response of Lavandula luisieri and Lavandula pedunculata chemotypes authors report a study concerning the anti-inflammatory potential of L. luisieri and L. pedunculata, assessed considering their chemical variability such as Essential oils
Reprot is good write and supèported by results.
I suggest minor revision reported below.
INTRODUCTION
Several industries have perceived this trend and are searching for bioactive products, biodegradable and non-toxic to humans and animals.
Please add reference that include the use of essential oils from plant as food prevent.
In this contex report paper such as (2021). Flavouring Extra-Virgin Olive Oil with Aromatic and Medicinal Plants Essential Oils Stabilizes Oleic Acid Composition during Photo-Oxidative Stress. Agriculture, 11(3), 266.
GC parameters as above; interface temperature: 250 ºC; MS source temperature
Please, report inject temperature for GC/MS analyses
Author Response
Dear Reviewer,
Please find enclosed our revised manuscript ‘The anti-inflammatory response of Lavandula luisieri and Lavandula pedunculata chemotypes’ now entitled ‘The anti-inflammatory response of Lavandula luisieri and Lavandula pedunculata essential oils’.
We thank you for the comments and suggestions, as they allowed us to improve the quality of our manuscript. We have revised it accordingly, highlighting all modifications in yellow. Moreover, an English revision was carried out as well as an improvement of the discussion section.
We truly hope that the revisions provided are sufficient to consider our work suitable for publication.
Reviewers suggestions:
In the paper The anti-inflammatory response of Lavandula luisieri and Lavandula pedunculata chemotypes authors report a study concerning the anti-inflammatory potential of L. luisieri and L. pedunculata, assessed considering their chemical variability such as Essential oils. Report is good write and supported by results.
I suggest minor revision reported below.
1) Introduction
- Several industries have perceived this trend and are searching for bioactive products, biodegradable and non-toxic to humans and animals. Please add reference that include the use of essential oils from plant as food prevent.
In this context report paper such as (2021). Flavouring Extra-Virgin Olive Oil with Aromatic and Medicinal Plants Essential Oils Stabilizes Oleic Acid Composition during Photo-Oxidative Stress. Agriculture, 11(3), 266.
We appreciate the suggestion of the reviewer. The reference was included in the introduction section.
- GC parameters as above; interface temperature: 250 ºC; MS source temperature
Please, report inject temperature for GC/MS analyses
We noticed that the inject temperature was already reported in the methods section (please see blue highlight).
Thank you in advance
Reviewer 2 Report
The authors present a research on two Lavandula species from Portugal, concerning the essential oils composition and the activity on some parameters involved in the inflammatory process.
In order to improve the manuscript, I have some observations and recommendations:
- The title could be improved
- The scientific name of the species is misspelled in some places in the text, please correct. After the first apparition in text of the whole name, the abbreviation should be used: luisieri and L. pedunculata
- If only the essential oil was used for testing, the term “extract” is not appropriate
- Introduction: the genus Lavandula includes several commercial products…. – please rephrase
- In order to define a chemotype according to the chemical composition of the essential oil, several determinations are required, and the concentration of the components should be expressed as average +/- standard deviation.
- The essential oil content from each vegetable product should be presented
- The discussions could be improved, being better directed towards the obtained results
- I recommend another 2 references in the subject, to be consulted:
Aprotosoaie AC, Gille E, Trifan A, Luca VS, Miron A. Essential oils of Lavandula genus: a systematic review of their chemistry. Phytochem Rev 2017;16:761–99.
Vairinhos, Jessica and Miguel, Maria Graça. "Essential oils of spontaneous species of the genus Lavandula from Portugal: a brief review" Zeitschrift für Naturforschung C, vol. 75, no. 7-8, 2020, pp. 233-245. https://doi.org/10.1515/znc-2020-0044
- English language needs improvements
Author Response
Dear Reviewer,
Please see attachment with our our revised manuscript ‘The anti-inflammatory response of Lavandula luisieri and Lavandula pedunculata chemotypes’ now entitled ‘The anti-inflammatory response of Lavandula luisieri and Lavandula pedunculata essential oils’.
We thank you for your comments and suggestions, as they allowed us to improve the quality of our manuscript. We have revised it accordingly, highlighting all modifications in yellow. Moreover, an English revision was carried out as suggested, as well as an improvement of the discussion section.
We truly hope that the revisions provided are sufficient to consider our work suitable for publication.
Thank you in advance

Round 2
Reviewer 2 Report
The manuscript was improved, almost all recommendations and observations were taken into account.